# Revisiting Pruning at Initialization through the Lens of Ramanujan Graph

**Duc Hoang, Shiwei Liu, Radu Marculescu & Zhangyang Wang**
Department of Electrical and Computer Engineering
University of Texas at Austin, Austin, TX 78712, USA
{hoangd,radum,atlaswang}@utexas.edu   shiwei.liu@austin.utexas.edu

## Abstract

Pruning neural networks at initialization (PaI) has received an upsurge of interest due to its end-to-end saving potential. PaI is able to find sparse subnetworks at initialization that can achieve comparable performance to the full networks. These methods can surpass the trivial baseline of random pruning but suffer from a significant performance gap compared to post-training pruning. Previous approaches firmly rely on weights, gradients, and sanity checks as primary signals when conducting PaI analysis. To better understand the underlying mechanism of PaI, we propose to interpret it through the lens of the *Ramanujan Graph* - a class of expander graphs that are sparse while being highly connected. It is often believed there should be a strong correlation between the Ramanujan graph and PaI since both are about finding sparse and well-connected neural networks. However, the finer-grained link relating highly sparse and connected networks to their relative performance (*i.e.*, ranking of difference sparse structures at the same specific global sparsity) is still missing. We observe that not only the Ramanujan property for sparse networks shows no significant relationship to PaI's relative performance, but maximizing it can also lead to the formation of pseudo-random graphs with no structural meanings. We reveal the underlying cause to be Ramanujan Graph's strong assumption on the upper bound of the largest nontrivial eigenvalue ($\hat{\mu}$) of layers belonging to highly sparse networks. We hence propose *Iterative Mean Difference of Bound* (IMDB) as a mean to relax the $\hat{\mu}$ upper bound. Likewise, we also show there exists a lower bound for $\hat{\mu}$, which we call *the Normalized Random Coefficient* (NaRC), that gives us an accurate assessment for when sparse but highly connected structure degenerates into naive randomness. Finally, we systematically analyze the behavior of various PaI methods and demonstrate the utility of our proposed metrics in characterizing PaI performance. We show that *subnetworks preserving better the IMDB property correlate higher in performance, while NaRC provides us with a possible mean to locate the region where highly connected, highly sparse, and non-trivial Ramanujan expanders exist.* Our code is available at: https://github.com/VITA-Group/ramanujan-on-pai.

## 1 Introduction

Deep neural networks (DNN) have demonstrated remarkable performance as they increase in size, i.e, test accuracy scales as a power law regarding model size and training data size (Hestness et al., 2017; Kaplan et al., 2020; Brown et al., 2020; Srivastava et al., 2022). Yet, the memory requirements and computational costs associated with the increased model size also grow prohibitively. Modern DNNs are widely recognized to be over-parameterized, and it has been shown that eliminating a significant number of parameters in a trained DNN does not compromise its performance (Han et al., 2015c; He et al., 2017). This over-parameterization property enables researchers to continually propose increasingly effective DNN pruning approaches that can dramatically shrink the model size while maintaining performance. The resultant sparse models can then be used with software and hardware that is optimized for sparsity, leading to faster training and inference.

Neural network pruning can generally be divided into three categories: post-training pruning (Mozer & Smolensky, 1989; Han et al., 2015a), during-training pruning (Gale et al., 2019; Louizos et al., 2018), and pre-training pruning (Lee et al., 2019; Wang et al., 2020), depending on the timing of

the pruning relative to the training phase. For instance, post-training pruning methods are generally effective when the primary goal is to reduce inference cost. However, these methods require training the dense model fully first, potentially multiple times if iterative pruning and retraining are used. With the prevalence of powerful foundation models such as GPT-3 (Brown et al., 2020), PaLM (Chowdhery et al., 2022), and DALL·E 2 (Ramesh et al., 2022), the prohibitively high cost of training these large models makes post-training pruning impractical. Therefore, pre-training pruning or pruning at initialization (PaI) is becoming increasingly attractive due to its potential to save time and resources end-to-end, by using a sparse DNN architecture from the outset.

The concept of pruning at initialization (PaI) was first introduced in SNIP (Lee et al., 2019), which removes the structurally unimportant connections at initialization via the proposed connection sensitivity. Follow-up works (Wang et al., 2020; Tanaka et al., 2020; Patil & Dovrolis, 2021) propose advanced pruning criteria to improve the performance of PaI. GraSP (Wang et al., 2020) aims to maintain the weights that can maximize the gradient flow. SynFlow (Tanaka et al., 2020) finds that previous PaI methods are prone to layer collapse and adopt iterative pruning to address it. Despite these advances, PaI still lags behind post-training pruning in terms of performance. A study by Frankle et al. (2021) suggests that connection ambiguity may explain the performance deficit, based on the finding that PaI exhibits surprising resilience against layer-wise random mask shuffling and weight re-initialization.

Prior works on PaI mainly focus on "training signals" such as gradient flow (Wang et al., 2020), layer collapse (Tanaka et al., 2020), or sanity check (random weight shuffling and re-initialization) (Frankle et al., 2021). Since the limited information (e.g., magnitude, gradient, and Hessian) that PaIs have access to can be very noisy (Frankle et al., 2020), pruning criteria based on such information are often ineffective. We conjecture that the graph topology of sparse neural networks, being relatively overlooked, can be an essential source of information for pruning at initialization. Graph theory has recently emerged as a particularly advantageous tool for analyzing DNN architectures. For example, You et al. (2020) offers a new efficient graph representation of DNN using *relation graphs* to formulate an efficient model generator. Liu et al. (2020) analyzes sparse neural networks with graph distance and shows a plenitude of sparse sub-networks with very different topologies while achieving similar performance. Vooturi et al. (2020); Pal et al. (2022); Bhardwaj et al. (2021); Prabhu et al. (2018) show that maximizing good graph connectivity, i.e. by maximizing a graph's expansion ratio, correlates to higher performance in hardware-efficient structured masks and lottery tickets (Frankle & Carbin, 2019). Unfortunately, prior efforts did not consider the pseudo-randomness that naturally emerges from very good expander graphs. Therefore by maximizing sparse graph connectivity, they are unwittingly prioritizing the formation of naive random graphs with no intrinsic structure meaning.

In this paper, we study PaI from the perspective of the Ramanujan bipartite graphs. The Ramanujan graph is a special graph in the bounded degree expander family, where the eigenbound is maximal (Nilli, 1991), thus leading to a maximum possible sparsity of a network while preserving the connectivity. The Ramanujan graph is intuitively well aligned with the main goal of PaI, i.e., finding sparse and well-connected neural networks. However, we find that there is still a missing link correlating the degree of connectivity to relative performance ranking at a particular sparsity. In addition, we also show that in situations where highly sparse and highly connected structures are demanded, it can be easy to generate pseudo-random graphs with no structural meanings unwittingly.

We reveal the underlying cause for such undesirable situations to be Ramanujan Graph's strong assumption on the upper bound of the largest nontrivial eigenvalue ($\hat{\mu}$) of layers belonging to highly sparse networks. We hence propose *Iterative Mean Difference of Bound* (**IMDB**) as a mean to relax $\hat{\mu}$ upper bound. Likewise, we also show there exists a lower bound for $\hat{\mu}$, which we call *the Normalized Random Coefficient* (**NaRC**), that gives us an accurate assessment for when sparse but highly connected structure deteriorates into randomness. Leveraging our (adjusted) Ramanujan graph-based framework, we then extensively investigate (1) whether the generated sparse structures by existing PaI approaches follow the Ramanujan characteristics, (2) if there exists a correlation between the Ramanujan graph property and the inference performance of the sparse structure, and (3) if the sparse structure is also a random graph. Ultimately, we aim to shed light on a new perspective on PAI effectiveness independent of weights, gradients, and losses.

Our contributions are summarized as follows:

- We are the first to reveal that the utility of the Ramanujan property is largely limited in analyzing irregular graphs at high sparsity, which is often the case in analyzing PaI-generated

architectures. We identify the root cause to be its strong assumption on the upper bound of the largest nontrivial eigenvalue $\hat{\mu}$ of the adjacency matrix and propose Iterative Mean Difference of Bound (IMDB) as an effective fix.

- We devise another novel metric to assess whether a Ramanujan graph is also random, which becomes increasingly likely with higher sparsity. We prove the existence of this metric by inferring from the definition of the expander mixing lemma. Our *NormAlized Random Coefficient* (NaRC) assesses the randomness of graphs, by characterizing the lower bound of $\hat{\mu}$ for which we can no longer distinguish expanders from randomly generated graphs.

- Our analysis shows that IMDB correlates strongly with the relative performance for different PaI's sparse masks at high sparsity. It further illustrates how NaRC can spot random structures, and provide interesting observations on the relationship between randomness and expansion.

## 2 RELATED WORK

Pruning methods (Mozer & Smolensky, 1989; LeCun et al., 1989; Hassibi et al., 1993; Molchanov et al., 2016; Han et al., 2015b) traditionally aim to remove the unnecessary components of DNNs, resulting in a subnetwork that can be efficiently deployed at inference. As the sizes of the modern DNNs have exploded, a vast amount of attention has been shifted to pruning them before training, targeting both training and inference efficiency. Lee et al. (2019) explicitly learn a connectivity importance score for weights and eliminate weights with the lowest scores. Wang et al. (2020) leverage the Hessian-gradient product to discover the importance of weight to the gradient flow. Iterative pruning approaches (Tanaka et al., 2020; de Jorge et al., 2021) show their efficacy to prevent layer collapse, ending up with pruned networks with very small width (Patil & Dovrolis, 2021). Although existing PaI methods surpass the naive baseline of random pruning, they are only able to identify useful layerwise sparsity levels rather than the specific weight patterns (Frankle et al., 2021; Su et al., 2020). As PaI only accesses very limited and noisy information (e.g., magnitude, gradient, and Hessian) from initialization, pruning criteria based on such information may not be effective.

On the other hand, the topology of sparse DNNs - the configuration of nodes and connections among them - can be another essential source of information. Mocanu et al. (2018) initialize sparse networks with a *Erdős-Rényi* graph and dynamically optimize the graph towards a scale-free network. Evci et al. (2020b) further expand the *Erdős-Rényi* graph to convolution neural networks, demonstrating large performance improvements. Liu et al. (2020) analyze sparse DNN with graph edit distance and show that there exists plenty of sparse sub-networks with distinct topologies that perform equally well. You et al. (2020) study the relationship between the graph structure and the neural networks from the *relation graphs* point of view. Vooturi et al. (2020); Pal et al. (2022); Bhardwaj et al. (2021) show that maximizing the graph connectivity of sparse networks correlates to higher performance in the structured masking and lottery tickets (Frankle & Carbin, 2019).

Recently, Ramanujan graphs have been linked to sparse structures (Pal et al., 2022; Vooturi et al., 2020) as naturally appealing criteria to produce sparse yet well-connected DNN models. We draw inspiration from two prior works: Vooturi et al. (2020) applies the Ramanujan bipartite graph products for efficient, structured sparsity. In short, they achieve run-time efficiency by decomposing a dense matrix into tiled multiplication and utilizing the Ramanujan principle to maximize connectivity between the decomposed tiled matrices. The more recent work by Pal et al. (2022) evaluates whether a lottery ticket's masks exhibit Ramanujan property. If they do not, then sparsity will repeatedly be halved until the generated masks are all Ramanujan graphs.

Besides the obvious difference that we uniquely work on pruning at initialization with end-to-end sparsity (no dense pre-training is involved), our work also differs from these prior arts in our tackling of irregular bi-graphs that arise from practical sparse DNNs: we are the first to identify a crucial limitation of Ramanujan property in analyzing irregular graphs under high sparsity, which seems to be overlooked by prior arts. We also highlight the danger of being overly expansive, which risks graphs deteriorating into randomness. We then propose ways to relax the upper constraint to mitigate the critical gap between theory and practice and a lower constraint to avoid the danger of randomness, which leads to a more rigorous analysis of sparse DNNs at initialization.

## 3 METHODOLOGY

Before we start, we want to explain some nomenclature that will be used excessively in our definitions. First, a "regular" graph refers to a graph where all its vertices have the same number of in/out edges; We refer to the number of edges as $d$. Analogously, when we talk about regular graphs, we refer to any dense DNN, such as Linear or Convolutional layers once. An "irregular" graph, on the other hand, refers to a graph with mixed number of edges for every vertex. Likewise, when speaking of irregular graphs, we refer to the resulting unstructured pruning of DNN layers with PaI. Later, we will explain how the DNN's layers are represented as graphs.

### 3.1 PRELIMINARY: BIPARTITE EXPANDER GRAPHS

In this work, we focus mainly on the bipartite case of expander graphs. However, before giving a formal definition, we state the following intuition: expander graphs are graphs where every subset of vertices is not "too large" and has "many connections" to other vertices that do not belong to the same subset. Typically, these graphs are regular; however, we shall extend the definition to consider **finite, connected, and irregular graphs** for our DNN analysis purpose.

**Definition 1.** A bipartite graph or bi-graph $G = (L \cup R, E)$ is a graph consisting of two disjoint sets of vertices $L$ and $R$ such that every edge from $E$ connects one vertex of $L$ and one vertex of $R$.

Many definitions will make use of $G = (V, E)$, where $V = L \cup R$ instead. For most cases, it is trivial to extend them to bipartite graphs. However, in case of extension ambiguity, we will clarify these definitions specifically for bi-graphs.

**Definition 2.** Let $S \subseteq V$ for $G = (V, E)$; we denote $N(S) = \{v \in V | \exists u \in S, (u, v) \in E\}$ to be the neighborhood set consisting of all adjacent vertices not in $|S|$.

Since we are working with bipartite graphs, $S \subseteq L$ and $N(S) \subseteq R$, and there are no edges between any two vertices in $S$.

**Definition 3.** A $(n, m, d, \gamma, \alpha)$-expander is a *d-left-regular*[1] bipartite graph $G = (L \cup R, E)$, where $|L| = n, |R| = m$ ($m \leq n$) and $\forall S \subseteq L$ s.t $|S| \leq \gamma \cdot n$ the neighborhood set of $S$ satisfies $|N(S)| \geq \alpha \cdot |S|$. Here $\gamma \in \{0, 1\}$ and $\alpha \in [0, d]$.

The $\alpha$ and $\gamma$ parameters control the expansion ratio of the expander and are dependent on one another. For example, by letting $\alpha = d$ the regularity of $L$, then $\gamma \leq \frac{1}{d}$ since if $|S| \geq \frac{n}{d}$, then we violate the neighborhood constraints namely $|N(S)| \geq \alpha \cdot |S| > n > m$. The expansion ratio, $\frac{|N(S)|}{S}$, is related to the Cheeger constant $h(G)$, whereby a small ratio signifies information bottleneck and a large $h(G)$ indicates the graph is strongly connected. A good bipartite expander graph should ensure a large Cheeger constant so that information can flow freely.

Let us denote $A \in \mathbb{R}^{|n+m| \times |n+m|}$ as the adjacency matrix of some $d$-regular bipartite graph $G = (L \cup R, E)$ with eigenvalues $\mu(G)$ s.t $\mu_0 \geq ... \geq \mu_{v-1}$, where $d = \mu_0 = |\mu_{v-1}|$, and corresponding eigenvectors $\phi$. We define $\hat{\mu}(G) = \max_{|\mu_i| \neq d} |\mu_i|$ to be the largest nontrivial eigenvalue. Due to the nature of bipartite graphs where $|\mu_0| = |\mu_1| = d$, we will often refer to $\hat{\mu}(G)$ as our third-largest eigenvalue. The expansion property of any $G$ is represented by its spectral gap $\mu_0 - \hat{\mu}(G)$ (see Hoory & Linial (2006)). A large gap indicates $G$ is more "spread out" which is the hallmark trait of the expander graph. Since $\mu_0 - \hat{\mu} \geq 0$, we need to determine a threshold on the spectral gap for which $G$ can be considered as a good expander. This brings us to the notion of Ramanujan graph property.

**Definition 4.** A *d-regular* graph is said to be a Ramanujan graph if $\hat{\mu}(G) \leq 2\sqrt{d-1}$, where $d$ is graph regularity. Alternatively, following the previous discussion, Ramanujan graph can also be expressed as $\hat{\mu}(G) \leq 2 * \sqrt{\mu_0 - 1}$, since $\mu_0 = d$.

Thus, all Ramanujan graphs are good expanders due to the convenient upper bound of $\hat{\mu}(G)$. However, after PaI **not all graphs are regular**. Therefore, we need to generalize **Definition 4** for all cases. To do so, we combine two inequalities: The first inequality states that the universal cover graph $\tilde{G}$ of $G$ satisfies $p(\tilde{G}) \geq 2 * \sqrt{d_{avg} - 1}$, where $p(\tilde{G})$ denotes the spectral radius of $\tilde{G}$ and $d_{avg}$ represents the average degrees of $G$ (for details see Hoory (2005)). The second inequality, following the results

---

[1]*d-left-regular* means all vertices in $L$ have $d$ number of edges.

of Hoory & Linial (2006), defines a graph to be Ramanujan iff $\hat{\mu}(G) \leq p(\tilde{G})$. We relate these two inequalities to form $\hat{\mu}(G) \leq 2 * \sqrt{d_{avg} - 1} \leq p(\tilde{G})$.

As a direct result, we can estimate any graph's expansion property as the difference between $\hat{\mu}(G)$ and its estimated Ramanujan's upper bound.

**Definition 5.** We denote the difference of bound as: $\Delta r = 2 * \sqrt{d_{avg} - 1} - \hat{\mu}(G)$ . A value $\Delta r < 0.0$ indicates a violation of the Ramanujan property and therefore the graph may not be a good expander.

In the case of bi-graphs, we rewrite **Definition 5** as $\Delta r = \sqrt{d_R - 1} + \sqrt{d_L - 1} - \hat{\mu}(G)$, where $d_R$ and $d_L$ are the average degree of $R$ and $L$ respectively. Additionally, because $\hat{\mu}(G)$ is the third largest eigenvalues by magnitude, a Ramanujan graph is only defined when $\min(d_L, d_R) \geq 3$.

### 3.2 ITERATIVE MEAN DIFFERENCE OF BOUND

In this subsection, we focus on discussing the value $\mu_0$ of irregular graphs, something we glossed over in the discussion of the previous subsection and it is also largely overlooked by prior works. Primarily, we want to address cases where $\mu_0 \neq d$ and assess the effect it can have on determining if a graph $G$ satisfies the Ramanujan property. Here, we explicitly define the general range of $\mu_0$ on all graphs, explain its negative effects on our analysis of high, and propose a way to mitigate the effects.

**Lemma 1.** The value of $\mu_0$ for any adjacency matrix $A$ is said to be $d_{avg} \leq \mu_0 \leq d_{max}$. If $G$ is a connected graph, $\mu_0 = d_{max}$ therefore $G$ is $d_{max}$-regular. We first prove $d_{avg} \leq \mu_0$ by using the Rayleigh quotient. We have:

$$\mu_0 = \max_{\mathbf{h} \in \mathbb{R}^n} \frac{\mathbf{h}^T A \mathbf{h}}{\mathbf{h}^T \mathbf{h}} \geq \frac{\mathbf{1}^T A \mathbf{1}}{\mathbf{1}^T \mathbf{1}} = \frac{\sum_{v \in V} d_v}{n} = d_{avg} \tag{1}$$

To prove $\mu_0 \leq d_{max}$, let $\phi_0$ be the corresponding eigenvector of $\mu_0$ and say that $v = \arg\max_u \phi(u)$. Without loss of generality, we have $\phi_0(v) \neq 0$. So, we show:

$$\mu_0 = \frac{(\mu_0 \phi_0)(v)}{\phi_0(v)} = \frac{(A\phi_0)(v)}{\phi_0(v)} = \frac{\sum_{(u,v) \in E} \phi_0(u)}{\phi_0(v)} \leq \sum_{(v,u) \in E} 1 = d_v \leq d_{max}. \tag{2}$$

Finally, we show that if $\mu_0 = d_{max}$, then $\frac{\sum_{(u,v) \in E} \phi_0(u)}{\phi_0(v)} = \sum_{(v,u) \in E} 1 = d_v = d_{max}$. In this way, $\phi_0$ is a constant vector with value $d_v$ for all vertices $u$ that are connected to $v$. By repeating the way $v$ was chosen then applied to $u$ for all such vertices $u$, we yield the result that $G$ is $d_{max}$-regular.

**$\Delta r$'s limitation for irregular graphs at high sparsity:** Now that we have proven the range of $\mu_0$, we can extend it to the following inequality: $\hat{\mu}(G) \leq 2 * \sqrt{d_{avg} - 1} \leq 2 * \sqrt{\mu_0 - 1}$. This inequality shows that Ramanujan's upper-bound estimation for $\hat{\mu}(G)$ is conservatively small.

Because of our interest in analyzing high sparsity (which is often the pursuit of PaI methods), $d_{max} - d_{avg}$ is possibly very large. By following $\Delta r$, we only qualify those graphs with tiny $\hat{\mu}(G)$ as Ramanujan graphs. We refer to the *expander mixing lemma* (see detail in Sauerwald & Sun (2011)), which relates to the relative degree of $\hat{\mu}(G)$ and the smaller $\hat{\mu}(G)$ (the more $G$ appears to be random). Consequently, in the irregular graph case, *we could dismiss valid expanders due to the overly limiting requirements while retaining only graphs with random sparse-graph structures when analyzing high sparsity*.

**IMDB.** We propose a new metric called Iterative Mean Difference of Bound ($\Delta r_{imdb}$) as a way to relax the $\Delta r$'s constraint. Let $G$ be our irregular graph, we define a set $K$ to be $\{K_i(V, E, d_i) \subset G | d_i \geq 3\}$ the set of $d$-regular subgraphs in $G$, $\Delta r_{imdb}$ is formally defined as:

$$\Delta r_{imdb} = \frac{1}{|K|} \sum_i^{|K|} (2 * \sqrt{d_i - 1} - \hat{\mu}(K_i)) \tag{3}$$

Since every graph in $K$ are regular, we do not have to estimate the upper bound of their $\hat{\mu}$. Intuitively, we say an irregular graph $G$ is a good expander if its regular subgraphs are good expanders. Finally, we extend Eq 3. for bi-graphs as:

$$\Delta r_{imdb} = \frac{1}{|K|} \sum_i^{|K|} (\sqrt{d_i - 1} + \sqrt{d_R - 1} - \hat{\mu}(K_i)) \tag{4}$$

where $d_R$ is the average degree of $R$ since we only care about $d$-left-regular bi-graphs for our settings. Our ablation study shows $\Delta r_{imdb}$ effectively correlates relative performance to relative connectivity for highly connected highly sparse structures.

### 3.3 NORMALIZED RANDOM COEFFICIENT

In the previous section, we mentioned the *expander mixing lemma* and its relation to $\hat{\mu}(G)$, but did not go into much detail about it. In this section, we define this relation and show how it can be used to prove the existence of our *NormAlized Random Coefficient* (NaRC).

**Definition 6.** Given a $d$-regular $G(V, E)$, the *expander mixing lemma* is given as:

$$||E(S,T)| - \frac{d|S||T|}{n}| \leq \mu_1(G)\sqrt{|S||T|} \tag{5}$$

where $S, T \subseteq V$, $S \cap T = \{0\}$, $n = |V|$, $\mu_1(G)$ is the second largest eigenvalue of $G$, and $|E(S,T)|$ is the total number of edges between $S$ and $T$. The intuition behind it is that the smaller $\mu_1(G)$, the more $G$ appears to be random. For more explanations, please refer to Sauerwald & Sun (2011).

For our irregular bi-graph case, we exchange $\mu_1$ and $\mu_0$ since they are equivalent, and replace $d$ with $d_L$, the average left degree. We can also exchange the definition of $S$ to that of **Definition 2**, with $S \subseteq L$, likewise replace $T$ with $N(S) \subseteq R$ and $n$ with $m = |R|$. The expression now reads as:

$$||E(S,N(S))| - \frac{d_L|S||N(S)|}{m}| \leq \mu_0(G)\sqrt{|S||N(S)|} \tag{6}$$

From **Lemma 1**, we know that $d_{avg}$ is the smallest possible value $\mu_0$ can be for $G$, thus the following inequality is true:

$$d_{avg} \leq \frac{||E(S,N(S))| - \frac{d_L|S||N(S)|}{m}|}{\sqrt{|S||N(S)|}} \leq \mu_0 \leq d_{max} \tag{7}$$

Eq 7 further highlights our earlier argument for IMDB, but we can now take it a step further. With this inequality, we can rewrite the Ramanujan upper bound to include the random expander graph subsets. This means:

$$\hat{\mu}(G) \leq 2\sqrt{\frac{||E(S,N(S))| - \frac{d_L|S||N(S)|}{m}|}{\sqrt{|S||N(S)|}} - 1} \tag{8}$$

By subtracting the right-hand side, we arrive at our definition for **NaRC**:

$$\sigma = (\frac{\hat{\mu}(G)^2}{4} + 1)\sqrt{|S| * |N(S)|} - ||E(S,N(S))| - \frac{d_L}{|m|}|S| * |N(S)|| \leq 0 \tag{9}$$

and we say that $G$ is a random expander if $\sigma \leq 0$, and $\frac{|\sigma|}{|L|*|R|}$ denote our normalized degree of randomness. From now on, when we refer $\sigma$, we will use its normalized version $\frac{\sigma}{|L|*|R|}$. From the above inequality, we can now clearly see how *expander mixing lemma* relates to $\hat{\mu}(G)$, and the smaller it is, the more random the graph becomes.

One reason we do not want our graphs to be reduced to randomness is that it conflicts with our desire for "specific connections" as stated by Frankle et al. (2021). Compared to trivial random sparsity, non-random and structurally meaningful masks are expected to provide an interpretation of where and how performance is derived. Our experiments show under the right circumstance, a network can learn to overcome randomness, and those with meaningful masks achieve significantly better performance on average.

### 3.4 BIPARTITE EXPANDER GRAPHS ON PRUNING AT INITIALIZATION

All models can be viewed as a sequence of computing graphs. Let $M$ denote the set of a $l$-layer model, we obtain $M = \{G_i, ..., G_l\}$ graphs, where $G_i = (L \cup R, E)$ is the $i$-th layer's graph representation with $L$ and $R$ indicates the input and output layer respectively. $M$ initially starts as a set of complete bipartite graphs. Pruning is then a process of edge sparsification on $M$ with the

resulting sub-graphs considered irregular expander graphs. The expander property appeals to network sparsification analysis because it can approximate a complete graph, as shown in Spielman (2018). However, approximating complete graphs using irregular graphs is still an open question. For our study, we consider only the convolution and linear layers of any given model.

**Convolution layer.** A convolutional weight consists of four dimensions namely the input channels, output channels, kernel width, and kernel height. To represent the convolution layer as a bipartite graph, we can unfold the input and kernels dimension to get $W \in \mathbb{R}^{|L| \times |R|}$, where $|L| = C_{in} * K_w * K_h$ and $|R| = C_{out}$. The resulting weighted graph is written as $G = (L \cup R, E, W)$. Note that the number of edges $|E| = |W|$.

**Linear Layer.** A linear weight consists of only the input and output channels, and bipartite conversion is trivial. We directly express its weight as $W \in \mathbb{R}^{|L| \times |R|}$, where $|L| = C_{in}$ and $|R| = C_{out}$. The resulting weighted graphs is similarly written as $G = (L \cup R, E, W)$.

**Prunning at initialization.** For each network, let $W$ denotes the set of weights $\{w_l \in \mathbb{R}^{n_l} | \forall l \in \{1, ..., L\}\}$ where $n_l$ is the number of parameters at layer $l$. Pruning is the process of generating binary masks $m_l \in \{0, 1\}^{n_l}$. A pruned subnetwork has weights $w_l \odot m_l$, where $\odot$ is the element-wise product. Most PaI methods have two stages: First, they issue scores $z_l \in \mathbb{R}^{n_l}$ to all weights. Second, they remove the score into mask $m_l$ with overall sparsity $s$. Pruning may occur iteratively or in one shot depending on the methods. We study the following representative PaI techniques:

• **Random** (Liu et al., 2022) is the most basic PaI method that uniformly prunes every layer with the same pruning ratio assigned globally. Each parameter is randomly assigned a score based on the normal distribution. • **ERK** (Evci et al., 2020a; Mocanu et al., 2018) initializes sparse networks with a *Erdős-Rényi* graph where small layers are usually allocated more weights. •**SNIP** (Lee et al., 2019) issues scores $s_l = |g_l \odot w_l|$ where $g_l$ and $w_l$ are gradients and weights respectively. The weights with the lowest scores after one iteration are pruned before training. •**GraSP** (Wang et al., 2020) removes weights that impeded gradient flows, by computing the Hessian-gradient product $h_l$ and issue scores $s_l = -w \odot h_l$. • **SynFlow** (Tanaka et al., 2020) iteratively prunes a model with its weights replaced with $|w_l|$. Each time, it propagates an input of 1's and computes the gradients based on the task's loss function $r_l$. It issues a score $s_l = |r_l \odot w_l|$ and removes the parameters with the smallest scores.

## 4 EXPERIMENTS

In this section, we examine our earlier claims with empirical results and see how they fare; furthermore, with supporting evidence, we answer questions regarding relationships between *Ramanujan to performance, randomness to performance, and Ramanujan to randomness*. Finally, we point out intuitions and what they imply for PaI under the lens of the Ramanujan perspective.

**Experimental settings.** We conduct our experiments with two different DNN architectures: Resnet-34 (He et al., 2016) and Vgg-16 (Simonyan & Zisserman, 2014) on CIFAR-10 (Krizhevsky, 2009). We run our experiments with three random seeds and initialize all PaI methods identically with three different initial weights generated by each seed to ensure fairness. Table 1 summarizes our standardized training configurations. We include additional results on CIFAR-100 in the Appendix.

Table 1: Summary of architectures and hyperparameters that we study in this paper.

| Model | Data | #Epoch | Batch Size | Optimizer | LR | LR Decay, Epoch | Weight Decay |
|---|---|---|---|---|---|---|---|
| Resnet-34 | CIFAR-10 | 250 | 256 | SGD | 0.1 | 10×, [160, 180] | 0.0005 |
| VGG-16 | CIFAR-10 | 250 | 256 | SGD | 0.1 | 10×, [160, 180] | 0.0005 |

### 4.1 OBSERVATIONS, INTUITIONS AND ABLATIONS

**Relationship between Ramanujan and performance:** In Figure 1, we confirm that Ramanujan $\Delta r$ indeed correlates with performance as a function of density (denser equates wider in general). However, this observation is already known (Pal et al., 2022), so it is not very exciting. The left column of Figure 1, which visualizes **Definition 5**, can only show that connectivity correlates strongly with density, but it cannot claim whether strong connectivity is related to *relative performance*. We attribute this missing link to the strong upper bound of $\hat{\mu}(G)$. While $\hat{\mu}(G)$ ensures that graphs that satisfy the inequality are expanders, we see that it holds little correlation with the actual performance potential of the network. This brings us to our first contribution, which is the **Iterative Mean Difference of Bound** $\Delta r_{imdb}$.

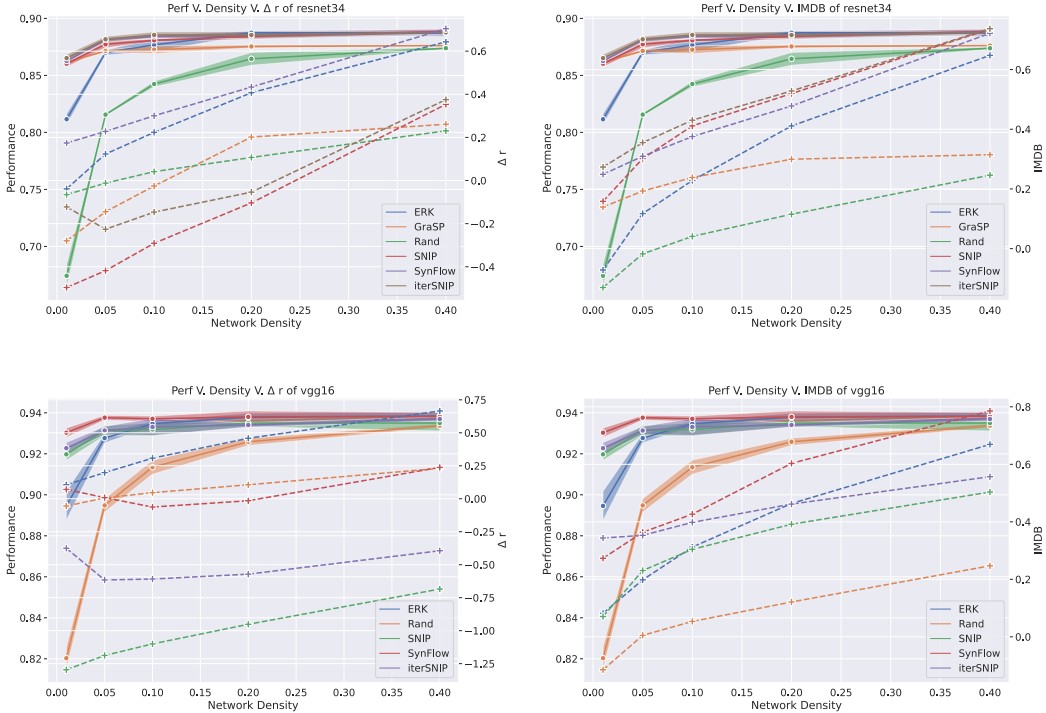

Figure 1: In these figures, solid lines always refer to the left y-axis, while dash lines refer to the right y-axis. Here, we illustrate the relationship between network density (x-axis), accuracy (left-y), and Ramanujan property ($\Delta r$ (left column) and $\Delta r_{imdb}$ (right column)) for Resnet-34 (top-row) and Vgg-16(bottom-row). First, we show in all cases Ramanujan graphs correlate strongly with performance (upward trend). Second, we show that $\Delta r_{imdb}$ also strongly correlates to relative performance between different PaI methods at different sparsity. Note that for Vgg-16, we dropped GrASP due to its inability to generate a feasible mask.

**Correlating relative performance with $\Delta r_{imdb}$:** Figure 1 right column shows that $\Delta r_{imdb}$ resolves the missing link stated early on. We demonstrate that $\Delta r_{imdb}$ is able to correlate the degree of connectivity of individual sparse structures with their final ranking in performance at specific density levels. While its effectiveness lessens with increasing density, it is undeniable that $\Delta r_{imdb}$ mirrors the performance trend of various sparse structures. Intuitively speaking, we can extend $\Delta r_{imdb}$ to be a performance ranker for unstructured sparse masks. Ultimately, the results prove our intuition that *an irregular graph $G$ is a good expander if its regular subgraphs are good expanders too.*

**On the relationship between performance and $\sigma$:** When we show that there exists a ratio $\sigma$ that can provide a randomness estimation on the graph, we are naturally curious about its relationship with performance. But first, we need to perform a sanity check on $\sigma$ to see if it works. We can confirm, using our two random methods (Rand and ERK) on Figure 3, that they always yield $\sigma \leq 0$, no matter the settings. Now we observe from the left column of Figure 3 that there seems to be little correlation between performance and the randomness of the graph at first glance. The lack of correlation would support previous observations made by Frankle et al. (2021); Liu et al. (2022) on the sufficiency of random pruning given the right sparsity. However, Figure 2 quickly dispels these notions, which neatly brings us to our next point.

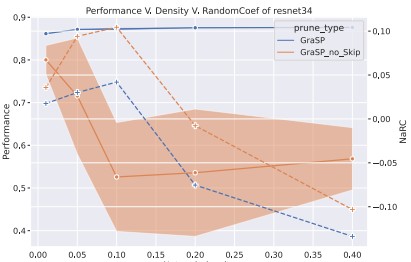

Figure 2: The relationship between performance and $\sigma$ for Resnet-34 with and without skip. We observe $\sigma$ starts to correlate with performance without skip-connections.

**The value of $\sigma$:** We observe an interesting inverse relationship between $\sigma$ and $\Delta r_{imdb}$ for Resnet-34 and Vgg-16. In Figure 3 right columns, we observe Resnet-34 to have a mutual correlation

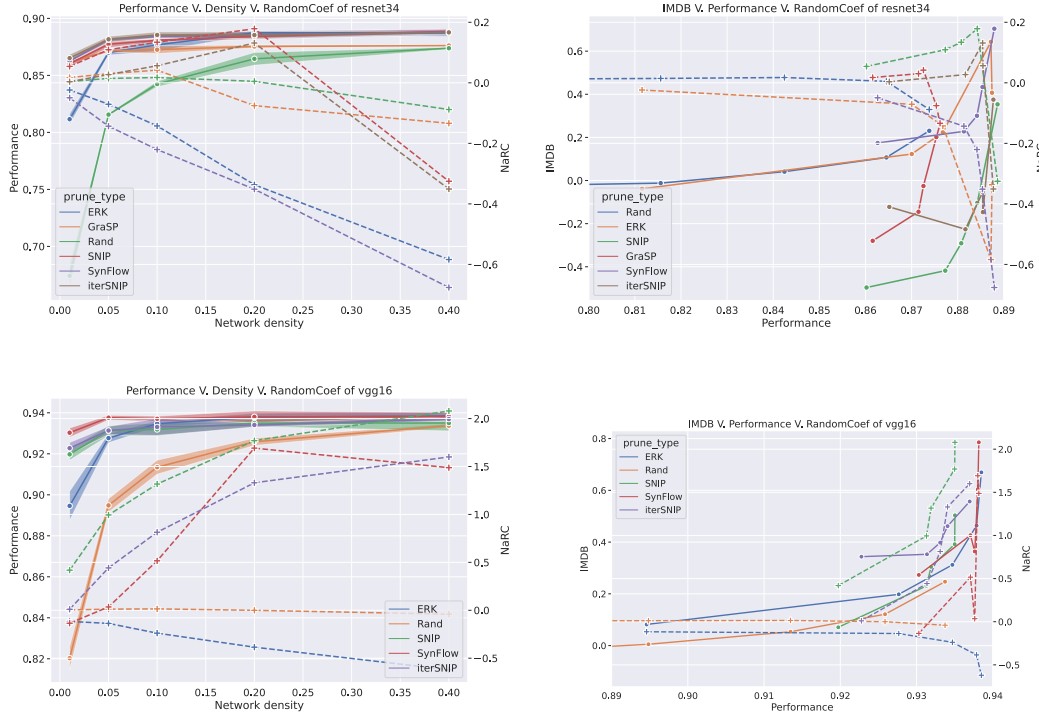

Figure 3: Identically formatted as Figure 1, on the left column, we illustrate the relationship between model's performance and network's randomness over global density for Resnet-34 and Vgg-16. On the right column, we try to correlate the Ramanujan characteristic ($\Delta r_{imdb}$) with our Normalized Random Coefficient (NaRC) over performance. The observation is interesting because the relationship are perfectly inverted between our two models.

between its degree of expansion and randomness that somehow yield increasingly better performance. Meanwhile, for Vgg-16, we see that a higher degree of expansion correlates to a lower degree of randomness, producing better performance. The second observation follows our specific intuitions, while the first observation seemingly contradicts them. How can randomness contribute to better performance? The answer turns out to be straightforward. In Figure 2, we compare the performance of Resnet-34 with and without skip connections at various densities. We observe that (1) as the model gets more sparse without skip connections, its masked structure becomes more specific (less random) to ensure gradient flows; (2) skip connections help carry information and act as a crutch to overcome the random nature of the model; (3) without skip connections, Resnet-34 becomes more specific; however, relying on gradients alone is not enough to recover the performance achieved by the model with skip connections. It all means that $\sigma$ negatively affects performance for both models, and skip connections can help alleviate the symptoms while exacerbating the problem as they effectively hide randomness.

Overall, we have shown a way to relax the strong constraint on $\hat{\mu}(G)$ such that the resulting graph's connectivity correlates strongly with its final performance. Further checking the lower bound for $\hat{\mu}(G)$ could indicate whether a sparse structure deteriorates into randomness. Tying all together, we now have the necessary tool to locate the region where highly connected, highly sparse, and non-trivial expanders exist. While not our primary objective, we foresee future PaI works utilizing our metrics as "checkers" to guide the design of their new criteria in selecting sparse structures.

## 5 CONCLUSION

This work delved into quantifying PaI from the perspective of Ramanujan graphs. Firstly, we introduced $\Delta r_{imdb}$ as a novel way to relax the strong upper bound of $\Delta r$ in cases of highly sparse, highly connected networks. Secondly, we proved that there existed a random coefficient called NaRC that could reliably estimate the degree of randomness for any given irregular sparse graph, which served as a lower bound for $\hat{\mu}(G)$ beyond which a sparse structure deteriorated into randomness. Ultimately, we provided a new perspective on the effectiveness of PAI that is independent of weights, gradients, and losses. This work was purely scientific and no negative impact was anticipated.

ACKNOWLEDGMENTS

The authors thank Peihao Wang for his valuable insights offered during the project discussions. Z. Wang is in part supported by NSF Scale-MoDL (award number: 2133861) and the NSF AI Institute for Foundations of Machine Learning (IFML).

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

# A ADDITIONAL EXPERIMENTS ON CIFAR-100

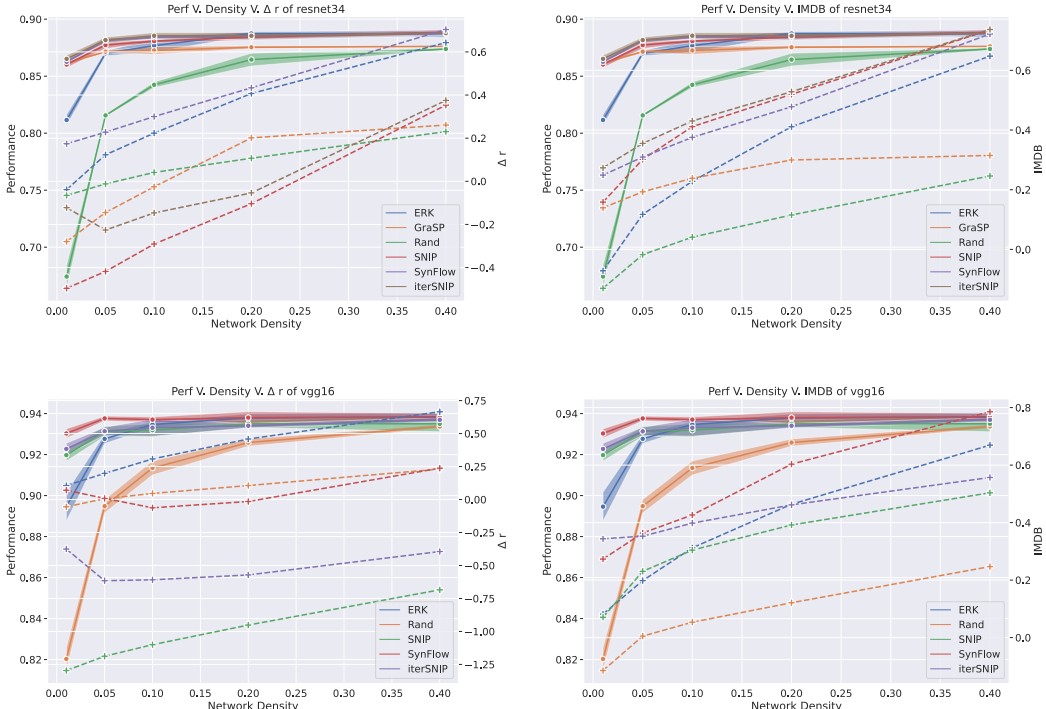

Figure 4: On these figures, solid lines always refer to the left y-axis, while dash lines refer to the right y-axis. Here we are illustrating the relationship between network density (x-axis), accuracy (left-y), and Ramanujan property ($\Delta r$ (left column) and $\Delta r_{imdb}$ (right column)) for Resnet-34 (top-row) and Vgg-16(bottom-row). First, we show in all cases Ramanujan graphs correlate strongly with performance (upward trend). Second, we show $\Delta r_{imdb}$ also strongly correlates to relative performance between different PaI methods at different sparsity.

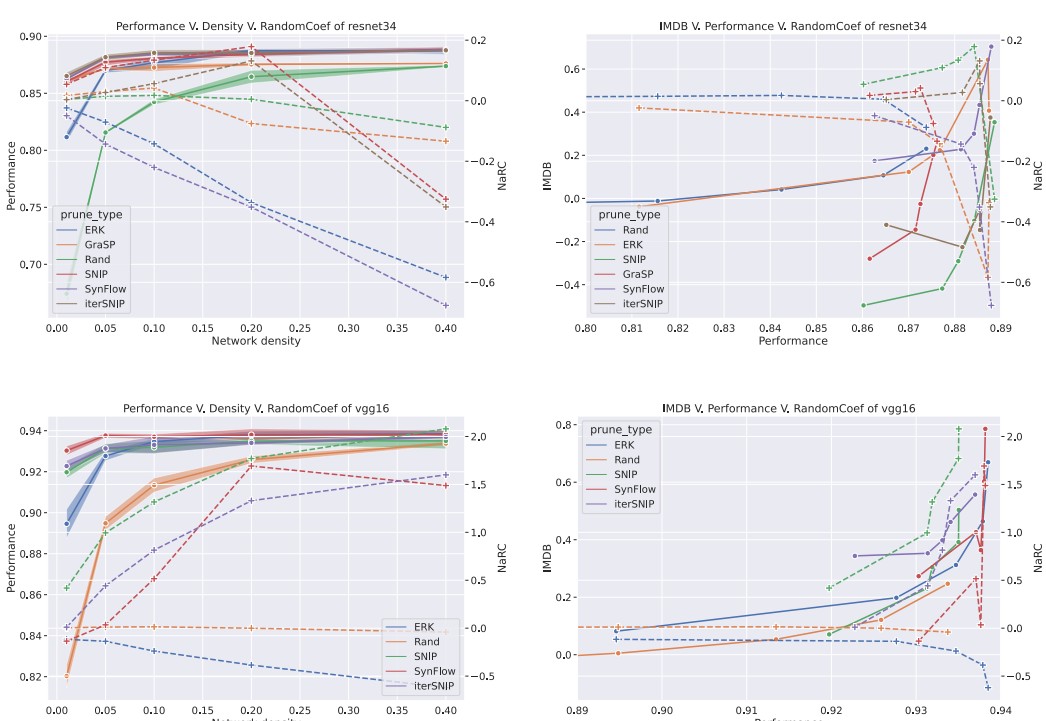

Figure 5: Identically formatted as Figure 1, on the left column, we illustrate the relationship between model's performance and network's randomness over global density for Resnet-34 and Vgg-16. On the right column, we try to correlate the Ramanujan characteristic ($\Delta r_{imdb}$) with our Normalized Random Coefficient (NaRC) over performance. The observation is interesting because the relationship are perfectly inverted between our two models.

