# OpenReview forum: "REVISITING PRUNING AT INITIALIZATION THROUGH THE LENS OF RAMANUJAN GRAPH"
_ICLR.cc/2023/Conference — ICLR 2023 notable top 5%_

### Official Review · Reviewer_pUbu · 2022-10-24

**Confidence:** 4
**Correctness:** 3
**Technical Novelty And Significance:** 3
**Empirical Novelty And Significance:** 2
**Recommendation:** 8

**Clarity, Quality, Novelty And Reproducibility:**

The paper is clearly written. The novelty is listed above. Experiments are straightforward (proof-of-concept level) and should be easy to reproduce.

**Strength And Weaknesses:**

Strength
1.	This paper went beyond intuition and builds a new mathematical framework for understanding PaI. The differences from previous PaI studies (gradient- and graph-based) are clearly elaborated.
2.	This paper also went beyond simply pulling off-the-shelf Ramanujan graph metrics to analyze DNN architectures (like prior art did). Instead it did several important modifications tailored for irregular bi-graphs and high sparsity levels, both are demanded in the PaI situation.
3.	The authors are the first to identify a crucial limitation of Ramanujan property in analyzing irregular graphs, i.e., we could dismiss valid expanders due to the overly limiting requirements while retaining only graphs with random sparse-graph structures when analyzing high sparsity. They proposed a new metric called Iterative Mean Difference of Bound (IMDB) to mitigate the gap. This finding could be of independent interest to other applications where Ramanujan graph is relevant.
4.	The authors also highlight the danger of being overly expansive, which risks graphs deteriorating into randomness. They proposed a lower constraint to avoid the danger of randomness.
5.	Overall, the authors show the key knob to be relaxing the strong upper bound constraint on the third large affinity matrix eigenvalue µ(G), such that the resulting graph’s connectivity correlates strongly with its final performance. Further checking the lower bound for µ(G) could indicate whether a sparse structure deteriorates into randomness. Experiments align well with their theories.

Weakness
1.	I did not follow what differences the authors were referring to, between “trivial random sparsity” and “structurally meaningful masks”. What’s the criteria?
2.	Experiments are relatively limited. For example, only CIFAR-10 dataset is used as the testbed. Will the observations scale up to larger datasets? Also, could the authors find out whether some PaI methods can be obvious winners under their proposed metrics?


**Summary Of The Paper:**

It is believed there should be a strong correlation between the Ramanujan graph and PaI since both are about finding sparse and well-connected neural networks. This paper digs deep and rigorously establishes their non-trivial relationships in fine-granularity, e.g., ranking of difference sparse structures at the same specific global sparsity.

**Summary Of The Review:**

See the strength and Weaknesses.

---

> ### Author Response · Authors · 2022-11-10
> **Responding to Reviewer pUbu**
>
> We thank reviewer pUbu for support and constructive feedback.
>
> **Q1**: I did not follow what differences the authors were referring to, between “trivial random sparsity” and “structurally meaningful masks”. What’s the criteria?
>
> The criteria for which we determine trivial random sparsity is implied in the Expander mixing Lemma where the probability for an edge to exist between any two nodes obeys the normal distribution. More specifically, we say a structure is random if its Normalized Random Coefficient is negative, and specific if it is positive. The magnitude of NaRC implies the relative strength of either assertion.
>
> **Q2**: Experiments are relatively limited. For example, only CIFAR-10 dataset is used as the testbed. Will the observations scale up to larger dataset
>
> |Dataset | Method | Sparsity | Acc | IMDB |
> |---|---|---|---|---|
> |vgg16|Rand|0.40|0.715067|0.239150|
> |vgg16|SNIP|0.40|0.726600|0.524375|
> |vgg16|iterSNIP|0.40|0.729167|0.571503|
> |vgg16|SynFlow|0.40|0.729200|0.772114|
> |vgg16|ERK|0.40|0.730233|0.668213|
> |resnet34|Rand|0.40|0.587233|0.243506|
> |resnet34|GraSP|0.40|0.599567|0.314578|
> |resnet34|iterSNIP|0.40|0.600833|0.739897|
> |resnet34|SNIP|0.40|0.602733|0.735337|
> |resnet34|ERK|0.40|0.612133|0.647162|
>
> |Dataset | Method | Sparsity | Acc | IMDB |
> |---|---|---|---|---|
> |vgg16|Rand|0.20|0.700667|0.111476|
> |vgg16|iterSNIP|0.20|0.722700|0.477686|
> |vgg16|SNIP|0.20|0.725067|0.409820|
> |vgg16|ERK|0.20|0.725600|0.461364|
> |vgg16|SynFlow|0.20|0.728100|0.587852|
> |resnet34|Rand|0.20|0.575567|0.112684|
> |resnet34|GraSP|0.20|0.599700|0.297880|
> |resnet34|iterSNIP|0.20|0.608100|0.534005|
> |resnet34|SNIP|0.20|0.603233|0.526346|
> |resnet34|ERK|0.20|0.612233|0.401588|
>
> |Dataset | Method | Sparsity | Acc | IMDB |
> |---|---|---|---|---|
> |vgg16|Rand|0.05|0.639567|-0.005361 |
> |vgg16|ERK|0.05|0.696500|0.193215 |
> |vgg16|iterSNIP|0.05|0.705767|0.361128 |
> |vgg16|SNIP|0.05|0.710500|0.243495 |
> |vgg16|SynFlow|0.05|0.731500|0.366998 |
> |resnet34|Rand|0.05|0.519567|-0.018530 |
> |resnet34|ERK|0.05|0.590933|0.101492 |
> |resnet34|GraSP|0.05|0.592900|0.188965 |
> |resnet34|SNIP|0.05|0.606167|0.297726 |
> |resnet34|iterSNIP|0.05|0.611033|0.352821 |
>
>
> |Dataset | Method | Sparsity | Acc | IMDB |
> |---|---|---|---|---|
> |vgg16|Rand|0.01|0.535033|-0.116971 |
> |vgg16|ERK|0.01|0.641700|0.028288 |
> |vgg16|iterSNIP|0.01|0.677200|0.354501 |
> |vgg16|SNIP|0.01|0.683600|0.063364 |
> |vgg16|SynFlow|0.01|0.703400|0.247729 |
> |resnet34|Rand|0.01|0.412833|-0.130412 |
> |resnet34|ERK|0.01|0.528167|-0.099985 |
> |resnet34|GraSP|0.01|0.583833|0.120330 |
> |resnet34|SNIP|0.01|0.587533|0.167307 |
> |resnet34|iterSNIP|0.01|0.596133|0.292046 |
>
>
> |Dataset | Method | Sparsity | Acc | IMDB |
> |---|---|---|---|---|
> |vgg16|Rand|0.10|0.675800|0.042206 |
> |vgg16|ERK|0.10|0.717400|0.308248 |
> |vgg16|iterSNIP|0.10|0.717900|0.407879 |
> |vgg16|SNIP|0.10|0.720200|0.325643 |
> |vgg16|SynFlow|0.10|0.730300|0.435706 |
> |resnet34|Rand|0.10|0.557033|0.036694 |
> |resnet34|GraSP|0.10|0.596567|0.237064 |
> |resnet34|SNIP|0.10|0.605067|0.416161 |
> |resnet34|ERK|0.10|0.606100|0.223561 |
> |resnet34|iterSNIP|0.10|0.607833|0.426809 |
>
> Here we include a performance/IMDB table for each PaI methods at different sparsity for CIFAR100. Reviewer can find the full results in the appendix of our revised paper. Overall we find the performance correlation between IMDB and performance remains strong between cifar10/100.
>
> **Q3**:  Also, could the authors find out whether some PaI methods can be obvious winners under their proposed metrics?
>
> Under extreme sparse ratio (between $1-10$\%), IterSNIP and Synflow are the top performers in both accuracy and expansion for Resnet34 and Vgg16 respectively under cifar10 and cifar100. Beyond 10\%, we are not as confident as weights start to play a more important role in the model's performance and IMDB loses its effectiveness. We made this observation in section 4.1 "Correlating relative performance with $\Delta r_{imdb}$"

---

### Official Review · Reviewer_wCJu · 2022-10-25

**Confidence:** 4
**Correctness:** 3
**Technical Novelty And Significance:** 3
**Empirical Novelty And Significance:** 3
**Recommendation:** 8

**Clarity, Quality, Novelty And Reproducibility:**

Clarity: very well written and easy to follow
Novelty: high (spotting important issues of Ramanujan Graph in analyzing PaI, which were overlooked before)
Reproducibility: seems good, but no code was submitted.


**Strength And Weaknesses:**


Strength
-	Most prior works on PaI focus on “training signals” such as gradients. This paper studies a complementary angle of graph theory
-	Although there exist prior work connecting Ramanujan Graph and PaI, they did not consider the pseudo-randomness and irregular bi-graphs in practical sparse NNs, and may prefer “collapsed” naïve random graphs (overly expansive)
-	The authors rigorously analyzed the limitation of Ramanujan Graph in modeling irregular graphs (practical DNNs) at high sparsity, and propose to relax the assumption on the eigenvalue upperbound
-	The authors further proposed another metric to detect and avoid “trivial randomness” at high sparsity, by characterizing the same eigenvalue’s lower bound.
-	Together, the authors demonstrate the novel finding, that a valid and desirable sparse NN architecture only exists when its graph adjacency matrix satisfies certain spectrum condition, e.g., its third-largest eigenvalue is both upper- and lower-bounded properly
-	The authors experimentally validated their findings on SOTA PaI methods including ERK, SNIP, and SynFlow.


Weakness
-	The paper does not construct any new PaI approach, only observing how PaIs are correlated with their graph properties. It would be great if the authors could briefly share future insights of improvements.
-	Only CIFAR-10 dataset is used in experiments. I’d suggest the authors to report on 1-2 more datasets in order to make their empirical evidences more consistently convincing.


**Summary Of The Paper:**

This paper proposes to interpret the (un)success of Pruning neural networks at initialization (PaI). through the lens of the Ramanujan Graph - a class of expander graphs that are sparse while being highly connected. Despite the intuitive belief that PaI and Ramanujan Graph should be naturally related, their underlying relationships are shown to be more sophisticated and non-trivial as the authors analyzed. The authors supposed their analysis with ample experiments.

**Summary Of The Review:**

Overall, I would tend to accept this paper. If the author can address my concerns. I would be more convinced.

---

> ### Author Response · Authors · 2022-11-10
> **Responding to Reviewer wCJu's comments**
>
> We thank reviewer wCju for support and constructive feedback.
>
> **Q1**: The paper does not construct any new PaI approach, only observing how PaIs are correlated with their graph properties. It would be great if the authors could briefly share future insights of improvements
>
> We have shown that from Ramanujan's perspective, structure relative "expansion-ness" implies relative performance. Suppose we view network pruning under the neural architectural search perspective, where the search space is the entire model's parameter size. In that case, we can apply the principle of our observation to reduce the search space into a much smaller subset while satisfying the Ramanujan expander property without any need for weights. We are working to address several open questions regarding structure generation and weight backfill for PaI application.
>
> **Q2**: Only CIFAR-10 dataset is used in experiments. I’d suggest the authors to report on 1-2 more datasets in order to make their empirical evidences more consistently convincing
>
>
> |Dataset | Method | Sparsity | Acc | IMDB |
> |---|---|---|---|---|
> |vgg16|Rand|0.40|0.715067|0.239150|
> |vgg16|SNIP|0.40|0.726600|0.524375|
> |vgg16|iterSNIP|0.40|0.729167|0.571503|
> |vgg16|SynFlow|0.40|0.729200|0.772114|
> |vgg16|ERK|0.40|0.730233|0.668213|
> |resnet34|Rand|0.40|0.587233|0.243506|
> |resnet34|GraSP|0.40|0.599567|0.314578|
> |resnet34|iterSNIP|0.40|0.600833|0.739897|
> |resnet34|SNIP|0.40|0.602733|0.735337|
> |resnet34|ERK|0.40|0.612133|0.647162|
>
> |Dataset | Method | Sparsity | Acc | IMDB |
> |---|---|---|---|---|
> |vgg16|Rand|0.20|0.700667|0.111476|
> |vgg16|iterSNIP|0.20|0.722700|0.477686|
> |vgg16|SNIP|0.20|0.725067|0.409820|
> |vgg16|ERK|0.20|0.725600|0.461364|
> |vgg16|SynFlow|0.20|0.728100|0.587852|
> |resnet34|Rand|0.20|0.575567|0.112684|
> |resnet34|GraSP|0.20|0.599700|0.297880|
> |resnet34|iterSNIP|0.20|0.608100|0.534005|
> |resnet34|SNIP|0.20|0.603233|0.526346|
> |resnet34|ERK|0.20|0.612233|0.401588|
>
> |Dataset | Method | Sparsity | Acc | IMDB |
> |---|---|---|---|---|
> |vgg16|Rand|0.05|0.639567|-0.005361 |
> |vgg16|ERK|0.05|0.696500|0.193215 |
> |vgg16|iterSNIP|0.05|0.705767|0.361128 |
> |vgg16|SNIP|0.05|0.710500|0.243495 |
> |vgg16|SynFlow|0.05|0.731500|0.366998 |
> |resnet34|Rand|0.05|0.519567|-0.018530 |
> |resnet34|ERK|0.05|0.590933|0.101492 |
> |resnet34|GraSP|0.05|0.592900|0.188965 |
> |resnet34|SNIP|0.05|0.606167|0.297726 |
> |resnet34|iterSNIP|0.05|0.611033|0.352821 |
>
>
> |Dataset | Method | Sparsity | Acc | IMDB |
> |---|---|---|---|---|
> |vgg16|Rand|0.01|0.535033|-0.116971 |
> |vgg16|ERK|0.01|0.641700|0.028288 |
> |vgg16|iterSNIP|0.01|0.677200|0.354501 |
> |vgg16|SNIP|0.01|0.683600|0.063364 |
> |vgg16|SynFlow|0.01|0.703400|0.247729 |
> |resnet34|Rand|0.01|0.412833|-0.130412 |
> |resnet34|ERK|0.01|0.528167|-0.099985 |
> |resnet34|GraSP|0.01|0.583833|0.120330 |
> |resnet34|SNIP|0.01|0.587533|0.167307 |
> |resnet34|iterSNIP|0.01|0.596133|0.292046 |
>
>
> |Dataset | Method | Sparsity | Acc | IMDB |
> |---|---|---|---|---|
> |vgg16|Rand|0.10|0.675800|0.042206 |
> |vgg16|ERK|0.10|0.717400|0.308248 |
> |vgg16|iterSNIP|0.10|0.717900|0.407879 |
> |vgg16|SNIP|0.10|0.720200|0.325643 |
> |vgg16|SynFlow|0.10|0.730300|0.435706 |
> |resnet34|Rand|0.10|0.557033|0.036694 |
> |resnet34|GraSP|0.10|0.596567|0.237064 |
> |resnet34|SNIP|0.10|0.605067|0.416161 |
> |resnet34|ERK|0.10|0.606100|0.223561 |
> |resnet34|iterSNIP|0.10|0.607833|0.426809 |
>
> Here we include a performance/IMDB table for each PaI methods at different sparsity for CIFAR100. Reviewer can find the full results in the appendix of our revised paper. Overall we find the performance correlation between IMDB and performance remain strong between cifar10/100

---

### Official Review · Reviewer_Bf71 · 2022-10-25

**Confidence:** 2
**Correctness:** 3
**Technical Novelty And Significance:** 2
**Empirical Novelty And Significance:** 3
**Recommendation:** 6

**Clarity, Quality, Novelty And Reproducibility:**

The paper provides a new perspective on pruning using the theory of spectral expanders, which are more general than simply using random graphs. The writing in some parts is confusing. For example, Lemma 1 reads "The value of $\mu_0$ for any adjacency matrix is said to be $d_{avg} \leq \mu_0 \leq d_{max}$." This looks like it's stating a definition even though it is part of the lemma statement, and there is no separation between the lemma statement and its proof.

**Strength And Weaknesses:**

General comments
-

- Why use the third largest eigenvalue of the adjacency matrix? Usually the spectral gap is taken to be the difference between the first and second largest eigenvalues, which is guaranteed to be positive if the graph is connected.

Strengths
-

- The idea of using expander graphs to guide pruning is interesting and unifies existing work.

Weaknesses
-

- The decision to not use existing measures of spectral expansion is not entirely convincing. You mention that the Ramanujan condition is too pessimistic for non-regular graphs when $d_{max}$ is used, but why not use something like the second eigenvalue of the normalized Laplacian?
- The decision to pass to all $K$-regular subgraphs seems somewhat arbitrary and possibly expensive to compute.
- The pruning measure proposed does not leverage the weights of the network at initialization, since it does not sparsify the network based
 on sensitivity of the loss function.


**Summary Of The Paper:**

This paper proposes a new method to prune neural networks at initialization. By analogy to previous work which used sparse random subnetworks to speed up training, this paper instead uses Ramanjuan graphs, which are well-studied and known to be favorable with respect to expansion. It relaxes the eigenvalue condition of Ramanujan graphs to a new property (IMDB) which empirically correlates to increased performance when the network is trained on image classification tasks.

**Summary Of The Review:**

The idea of measuring performance of pruning in terms of spectral properties of the adjacency matrix is a good idea. However, the choices made in deciding upon the IMDB measure seem arbitrary in comparison to existing measures of spectral expansion. The choice of comparing to $\delta r$ is not entirely fair as a baseline, since that particular bound was designed around regular graphs.

---

> ### Author Response · Authors · 2022-11-10
> **Responding to Reviewer Bf71's comments part 1**
>
> We thank reviewer Bf71 for their constructive feedback. We hope our responses adequately address the concerns regarding the novelty of our work.
>
> First, we would like to clarify a misunderstanding reviewer might have had about our paper. In this paper, we \textbf{do not} propose new pruning criteria, instead, we provide a change in the Ramanujan perspective on irregular sparse structures that strongly correlate expansion property (without weights) and relative final performance at a specific sparsity.
> Our primary contributions are:
> 1. Discovering the limitation of current Ramanujan in PaI.
> 2. Fix by proposing Iterative Mean Difference of Bound (IMDB) to relax of Ramanujan's upper-bound.
> 3. Make a connection between randomness and PaI and propose a randomness check with Normalized Random Coefficient.
>
> We will make sure to make this point clearer in our write-up.
>
> Now, we will address all reviewer's concerns on a point-by-point basis. To help manage our answers, we define several notations here.
> We denote $\mu_1$ and $\mu_2$ as the second and third eigenvalues of a bipartite adjacency matrix $A$, respectively. $d_L$ and $d_R$ are the left and right degrees of a regular bipartite graph.
>     $\mathcal{L}$ is the normalized Laplacian.
>
> **Q1**: Why use the third largest eigenvalue of the adjacency matrix? Usually the spectral gap is taken to be the difference between the first and second largest eigenvalues, which is guaranteed to be positive if the graph is connected
>
>  We offer theoretical and practical answers:
>
> 1. For the theoretical answer, we refer the reviewer to four previous fundamental works on bipartite graphs [1, 2, 3, 4]. [1,2] proved that for every $\epsilon > 0$, the second largest eigenvalue $\mu_1$ satisfies $\mu_1 \geq \sqrt{d_L -1}+ \sqrt{d_R -1} - \epsilon$, where $d_L$ and $d_R$ are the left and right degree of a regular bipartite graph respectively. [3] shows that $\mu_1$ cannot be asymptotically smaller than the right limit of the asymptotic support for the eigenvalue distribution, which is $\sqrt{d_L -1}+ \sqrt{d_R -1}$. Therefore according to [1,2,3], a bipartite graph cannot be an expander if we consider $\mu_1$ since it violates Definition 4 in section 3.1. Conveniently , [4] states that the third eigenvalue $\mu_2 \leq \sqrt{d_L - 1} + \sqrt{d_R -1} + \epsilon$ with high probability. Therefore, to extend bipartite graphs to Ramanujan graphs, we consider its $\mu_2$ instead. This fundamental observation is also made by [5,6].
>
> 2. For the practical answer, we direct the reviewer to the definition of $\hat{\mu}$ in section 3.1; we defined $\hat{\mu}(G) = \max_{|\mu_i| \not = d } |\mu_i|$ to be a maximum function over all nontrivial eigenvalues, where we emphasize $\mu_0 = \mu_1 = d$ are trivial values. Because $A$, our binary adjacency matrix, is symmetric; its eigenvalues satisfy  $\mu_0 \geq \mu_1 \geq \mu_2 \geq ... \geq \mu_{n+m-1}$ and $|\mu_0| = |\mu_{n+m-1}|$($n$ and $m$ are the total left and right nodes); therefore we directly relate $\hat{\mu}(G)$ with $\mu_2$. We will make this point clearer in our paper.
>
>
>
>  [1] Keqin Feng and Wen-Ching Winnie Li. Spectra of Hypergraphs and Applications. Journal of Number Theory, 60(1):1–22, September 1996. ISSN 0022-314X. doi: 10.1006/jnth.1996.0109
>
>  [2] Wen-Ching Winnie Li and Patrick Sole. Spectra of Regular Graphs and Hypergraphs and Orthogonal Polynomials. European Journal of Combinatorics, 17(5):461–477, July 1996. ISSN 0195-6698. doi: 10.1006/eujc.1996.0040.
>
>  [3] C. D. Godsil and B. Mohar. Walk generating functions and spectral measures of infinite graphs. Linear Algebra and its Applications, 107(Supplement C):191–206, August 1988. ISSN 0024-3795. doi: 10.1016/0024-3795(88)90245-5.
>
>  [4] Brito, G., Dumitriu, I., & Harris, K. (2022). Spectral gap in random bipartite biregular graphs and applications. Combinatorics, Probability and Computing, 31(2), 229-267. doi:10.1017/S0963548321000249
>
> [5] https://sites.math.washington.edu/~morrow/papers/chris-thesis.pdf
>
> [6] Pal, Bithika et al. “A Study on the Ramanujan Graph Property of Winning Lottery Tickets.” ICML (2022).

---

> > ### Author Response · Authors · 2022-11-10
> > **Responding to Reviewer Bf71's comments part 2**
> >
> > **Q2**: The decision to not use existing measures of spectral expansion is not entirely convincing. You mention that the Ramanujan condition is too pessimistic for non-regular graphs when $d_{max}$  is used, but why not use something like the second eigenvalue of the normalized Laplacian?
> >
> > We assume the reviewer meant $d_{avg}$ instead of $d_{max}$, since letting $d_{max}$ be the Ramanujan upper-bound is overly optimistic for which all sparse structures will be qualified as Ramanujan expanders, and it is also not what we referred to in our paper. We notice that the Reviewer questions why we do not use the normalized Laplacian ($\mathcal{L}$) to decompose our eigenvalues. This is because the normalized Laplacian of a symmetrical adjacency matrix is $\mathcal{L} = I - D^{\frac{-1}{2}}AD^{\frac{-1}{2}}$ which is simply a scaled and shifted version of the adjacency matrix where $D^{\frac{-1}{2}}AD^{\frac{-1}{2}}$ is $\frac{1}{\sqrt{d_L*d_R}}A$. Therefore a spectral gap for $A$ implies that one exists for $\mathcal{L}$. By using $A$, we simplified our notations and aligned our paper's objective with the well-known definition of Ramanujan. As for why we don't use the second eigenvalue, we refer to our previous answer.
> >
> > **Q3**: The decision to pass to all K-regular subgraphs seems somewhat arbitrary and possibly expensive to compute.
> >
> > The reason for scanning for K-regular structures within an irregular graph is explained in depth in section 3.2. To paraphrase, we wish to avoid scenarios where we could easily dismiss suitable expanders (those with negative bound differences) due primarily to the overly strict Ramanujan criteria while qualifying only naive random sparse-graph when analyzing high sparsity. As for computing complexity, this is a trivial problem of left-hand node subset identification with K-out degrees and their associated right-hand nodes; because K-regular graphs are highly sparse, calculating for their $\mu_2$ is relatively inexpensive.
> >
> >  **Q4** : The pruning measure proposed does not leverage the weights of the network at initialization, since it does not sparsify the network based on sensitivity of the loss function.
> >
> > To the Reviewer's attention again, we \textbf{do not} propose new pruning criteria;  instead, our goal is to examine current PaI methods (which did not consider topology in their methods) from the Ramanujan perspective - and we never claimed this was the only relevant perspective. We did not consider the perspectives of weight or sensitivity, simply because current PaI methods were all built on top of them already.  Concretely, we identified important issues in the current Ramanujan perspective on irregular sparse structures that strongly correlate expansion property (without weights) and relative final performance at a specific sparsity. Our primary contributions are re-iterated as:
> >         1. Discovering the limitation of current Ramanujan in PaI.
> >         2.  Fix by proposing Iterative Mean Difference of Bound (IMDB) to relax of Ramanujan's upper-bound.
> >         3. Make a connection between randomness and PaI and propose a randomness check with Normalized Random Coefficient.
> >
> > **Q5**: The choice of comparing to $\Delta r$ is not entirely fair as a baseline, since that particular bound was designed around regular graphs.
> >
> > We offer several insights regarding the usage of as the baseline. First, $\Delta r$ is not the Ramanujan bound but is the distance from the bound. Second, we refer to section 3.1, between Definition 4, and Definition 5 on how the Ramanujan upper bound has been extended to the irregular graph by using the average degree. Finally, our contribution is directly about how this universally accepted extension by prior works is actually too draconian (meaning using it may prioritize only random structures and exclude valid expanders). Therefore, we believe using $\Delta r$ is an entirely fair comparison for our contribution. For more detail, we refer the reviewer to sections 3.2 and 3.3.

---

> > > ### Comment · Reviewer_Bf71 · 2022-11-21
> > > **Reply to revisions**
> > >
> > > Thank you for your detailed response and for clearing up some of my confusion and concerns. I'm now convinced that your criteria are well-motivated and provide a valuable new perspective on pruning at initialization and its relation to graph connectivity measures, as well as on identifying important subnetworks for training. I will increase my score to 6.

---

### Decision · Program_Chairs · 2023-01-20

**Decision:**

Accept: notable-top-5%

**Justification For Why Not Higher Score:**

N/A

**Justification For Why Not Lower Score:**

From a rigorous graph theory perspective, the authors revised the fundamental limitation of Ramanujan Graph in modeling irregular graphs at high sparsity. They proposed two fixes, i.e., relaxing the assumption on the eigenvalue upper-bound, and meanwhile characterizing the same eigenvalue’s lower bound. Together, the authors demonstrate that a good sparse NN exists only when the spectrum of its graph adjacency matrix satisfies the upper- and lower-bounded conditions on the third-largest eigenvalue.

Initially the experiments were only demonstrated on CIFAR-10, which some reviewers expressed concerns about. But after rebuttal the authors presented more experiments which add to satisfactory empirical credibility.


**Metareview: Summary, Strengths And Weaknesses:**

This is a very compelling paper that provides a valuable new perspective on pruning at initialization and its relation to graph connectivity measures. In particular, it proposes to interpret the (un)success of pruning neural networks at initialization (PaI) through the lens of the Ramanujan Graph – a renowned class of expander graphs that are sparse while being highly connected. While connecting expansive graph theory to NN architectures has been briefly studied before, this paper extended the depth of study much, by taking the pseudo-randomness and irregular bi-graphs in practical sparse NNs into account for the first time, as those may prefer “collapsed” naïve random or overly expansive graphs. After rebuttal discussions, all reviewers unanimously offer their strong support, and the AC fully echoes. Hence, it clearly makes a strong acceptance case.

**Note From Pc:**

if the above contains the word "oral" or "spotlight" please see: "oral" presentation means -> notable-top-5% and "spotlight" means -> notable-top-25%. As stated in our emails, we are disassociating presentation type from AC recommendations

**Summary Of Ac-Reviewer Meeting:**

N/A